# Electrosynthesis of ethylene glycol from $C_1$ feedstocks in a flow electrolyzer

Rong Xia [1,2], Ruoyu Wang[3], Bjorn Hasa [2], Ahryeon Lee[2], Yuanyue Liu [3] ✉, Xinbin Ma [1] ✉ & Feng Jiao [2] ✉

Ethylene glycol is a widely utilized commodity chemical, the production of which accounts for over 46 million tons of $CO_2$ emission annually. Here we report a paired electrocatalytic approach for ethylene glycol production from methanol. Carbon catalysts are effective in reducing formaldehyde into ethylene glycol with a 92% Faradaic efficiency, whereas Pt catalysts at the anode enable formaldehyde production through methanol partial oxidation with a 75% Faradaic efficiency. With a membrane-electrode assembly configuration, we show the feasibility of ethylene glycol electrosynthesis from methanol in a single electrolyzer. The electrolyzer operates a full cell voltage of 3.2 V at a current density of 100 mA cm$^{-2}$, with a 60% reduction in energy consumption. Further investigations, using operando flow electrolyzer mass spectroscopy, isotopic labeling, and density functional theory (DFT) calculations, indicate that the desorption of a *$CH_2OH$ intermediate is the crucial step in determining the selectively towards ethylene glycol over methanol.

Ethylene glycol, with a global capacity of 42 Mt/yr, is widely used as an antifreeze and precursor of polyethylene terephthalate (PET, the fourth-most produced synthetic plastic in the world)[1,2]. Conventionally, ethylene glycol is produced through ethylene partial oxidation at 220–280 °C and 1–3 MPa, followed by ethylene oxide hydrolysis (Supplementary Fig. S1)[3–5]. Current technology faces two major challenges: overoxidation of ethylene resulting in a significant amount of $CO_2$ emission, and hydrolysis of ethylene oxide forming higher homologues (i.e., diethylene glycol, triethylene glycol, and tetraethylene glycol)[2,6]. Alternatively, ethylene glycol can be produced from $C_1$ feedstocks, such as methanol and syngas[7–9]. While the alternative process avoids the hydrolysis of ethylene oxide, enabling a selective production of mono-ethylene glycol, nitric oxides are used to facilitate the C-C coupling step, which leads to significant $NO_x$ emissions[10,11]. Regardless of the feedstocks, current industrial processes for ethylene glycol production are accompanied by a substantial carbon footprint, 1.1 t$CO_2$ (ethylene) and 3.1 t$CO_2$ ($C_1$) per ton of ethylene glycol produced[1,12].

Electrosynthesis of ethylene glycol driven by renewable electricity can potentially reduce $CO_2$ emission[1,13–15]. Ethylene glycol electrosynthesis has been reported through ethylene electrooxidation on AuPd and $TiO_2$-$RuO_2$ catalysts[1,16]. However, the undesired side reactions, especially full oxidation of ethylene to $CO_2$, limit the operating current density. The ethylene glycol Faradaic efficiency is observed to decrease from 65% to 0.5% when the current density is above 10 mA cm$^{-2}$, which is far from commercial applications at their current state[16]. To circumvent the issues associated with ethylene overoxidation and slow reaction rate, an alternative pathway for ethylene glycol electrosynthesis from $C_1$ feedstocks is attractive because it avoids ethylene oxide as the key intermediate[7,8]. Formaldehyde has shown the potential of dimerization to produce ethylene glycol, however making ethylene glycol from formaldehyde is not economically competitive[17,18]. In order to make one ton of ethylene glycol, the total cost of feedstock using the formaldehyde is USD 1176 (Supplementary Table S1 and Supplementary Fig. S2), higher than the market price of ethylene glycol (USD 838). While the cost of raw material is only USD 361 via the methanol route, 70% lower than the

[1]Key Laboratory for Green Chemical Technology, School of Chemical Engineering and Technology, Tianjin University, Tianjin 300072, China. [2]Center for Catalytic Science and Technology, Department of Chemical and Biomolecular Engineering, University of Delaware, Newark, DE 19716, USA. [3]Texas Materials Institute and Department of Mechanical Engineering, The University of Texas at Austin, Austin, TX 78712, USA. ✉e-mail: yuanyue.liu@austin.utexas.edu; xbma@tju.edu.cn; jiao@udel.edu

formaldehdye approach and is ~40% lower than the conventional ethylene pathway (USD 557). However, there is no report on electrosynthesis of ethylene glycol from methanol and the key challenge is to maintain high energy efficiency at industrial-level current density.

Herein, we show an electrocatalytic synthesis of ethylene glycol from methanol in a single electrolyzer (Fig. 1). The redox reaction is divided into an oxidation reaction (methanol to formaldehyde) and a reduction reaction (formaldehyde to ethylene glycol). At the cathode, formaldehyde is reduced on carbon-based catalysts to form ethylene glycol. At the anode, methanol partial oxidation is performed using platinum-based catalysts and methanol is dehydrogenated to formaldehyde. A membrane electrode assembly-based electrolyzer was constructed, which substantially reduced the cell voltage and thus improved the energetic efficiency of ethylene glycol production. The reaction mechanism is further investigated via operando flow electrolyzer mass spectroscopy, isotopic labeling experiments, and density functional theory calculations, with a specific focus on the reaction mechanism of C-C coupling towards ethylene glycol formation.

## Results and discussion
### Formaldehyde electroreduction to ethylene glycol

In the proposed reaction pathway for ethylene glycol electrosynthesis, formaldehyde functions as an intermediate to facilitate C-C coupling towards ethylene glycol, while formaldehyde electroreduction to methanol is the undesired side reaction in this process. Hence, shifting formaldehyde electroreduction to ethylene glycol versus methanol is the key challenge. We first conducted catalyst screening for formaldehyde electrochemical reduction under a constant current ranging from 10 mA cm$^{-2}$ to 200 mA cm$^{-2}$ on various catalysts. Metal nanoparticles, including cobalt, nickel, silver, copper, and palladium were supported on titanium felt and carbon black was loaded on porous carbon paper. Only hydrogen, methanol, and ethylene glycol are observed as the major products of formaldehyde reduction. Figure 2a summarizes the performance of all the catalysts under a constant current of 25 mA cm$^{-2}$. Based on the selectivity, catalysts can be divided into three categories: cobalt and nickel mostly favor hydrogen evolution reaction; silver, copper, and palladium selectively catalyze the formaldehyde hydrogenation to methanol; carbon is the only catalyst that favors the C-C coupling in a wide potential range with a Faradaic efficiency of ethylene glycol up to 47% (Supplementary Fig. S3). The carbon source of ethylene glycol is examined with nuclear magnetic resonance (NMR) and gas chromatography–mass spectrometry (GC-MS) using $^{13}$C-labeled formaldehyde ($^{13}CH_2O$) as starting material and carbon black as catalysts. The $^{13}$C and $^{1}$H NMR spectra confirmed that $^{13}$C-labeled ethylene glycol is the product of formaldehyde reduction (Supplementary Fig. S4). Compared with ethylene glycol generated from non-labeled formaldehyde, the ethylene glycol synthesized using $^{13}CH_2O$ feedstocks exhibited an apparent 2 amu mass shift (Fig. 2b), suggesting that both carbon atoms were originated from $^{13}CH_2O$ (See detailed discussion in supporting information).

The active site of carbon-based material is investigated to better understand the unique performance of carbon catalysts in formaldehyde electrochemical reduction. Commercial carbon black was chosen due to its low cost and high surface area. Prior to the test, carbon black is cleaned with hydrochloric acid and washed with deionized water, followed by calcination at 500 °C for 3 h under an Ar atmosphere to remove any contaminant. The X-ray photoelectron spectroscopy (XPS) characterization reveals that there is trace amount of oxygen-containing functional groups on the surface as shown in the schematics in Fig. 2c. The oxygen-containing functional group in carbon black catalysts can be reduced by thermal treatment at elevated temperatures. The relationship between temperature and oxygen content is depicted in Supplementary Fig. S5. The results show that by eliminating oxygen functional groups via thermal treatment, the Faradaic efficiency (FE) for ethylene glycol improved, and side reactions were minimized (See detailed discussion in supporting information). However, extended annealing at higher temperatures was unable to remove additional oxygen functional groups and led to a degradation in the pore properties and surface area of the carbon black, as documented in earlier research[19]. The impact of oxygen-containing functional group is further investigated by preparing oxygen-doped carbon black through chemical oxidation method and electrochemical oxidation method, denoted as Chem-O-carbon black and Electro-O-carbon black, respectively. The chemically oxidized carbon black is prepared by refluxing carbon black in 70 wt% nitric acid at 70 °C for 12 hours. The electrochemically oxidized carbon black is made through cyclic voltametric oxidation of carbon black in 0.5 M sulfuric acid. The XPS spectra suggests that chemical oxidation mainly improves the carboxylic groups and carbonyl groups on the surface while the electrochemical oxidation method is beneficial to increase the content of hydroxyl groups (Fig. 2d). The as-prepared oxygen-doped carbon catalysts are tested in formaldehyde electroreduction. Compared with carbon black, oxygen-doped carbon black has a significantly lower ethylene glycol FE. The maximum ethylene glycol FE is 41.6% on Chem-O-carbon black and 16.7% on Electro-O-carbon black. Both oxygen-doped carbon catalysts exhibit an ethylene glycol FE of <7% at a current density of 200 mA cm$^{-2}$ (Fig. 2e). The impact of oxygen-containing functional groups is further investigated by varying the content of oxygen-containing functional groups. Carbon black is refluxed in 70 wt% nitric acid at 70 °C for 3 h and 12 h, denoted as Chem-O-carbon black-3 h and Chem-O-carbon black-12 h and the oxygen content increases from 8.9% to 14.4%. The two HNO$_3$-treated carbon black were tested

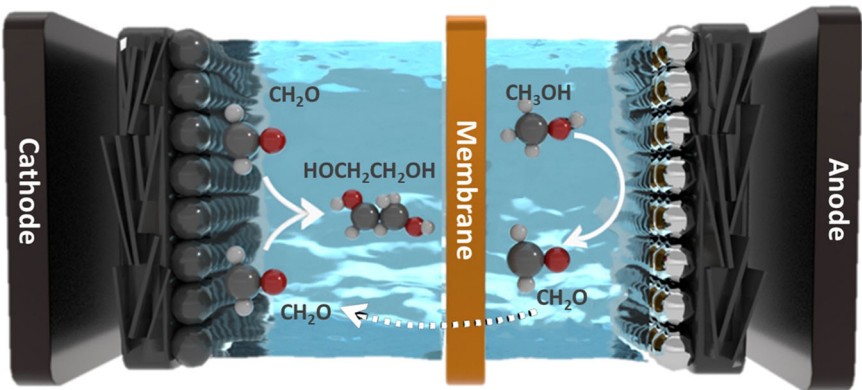

**Fig. 1 | Schematics of ethylene glycol electrosynthesis from methanol through a redox mechanism.** Methanol partial oxidation is performed using platinum-based catalysts to produce formaldehyde at the anode and formaldehyde is reduced on carbon-based catalysts to form ethylene glycol at the cathode.

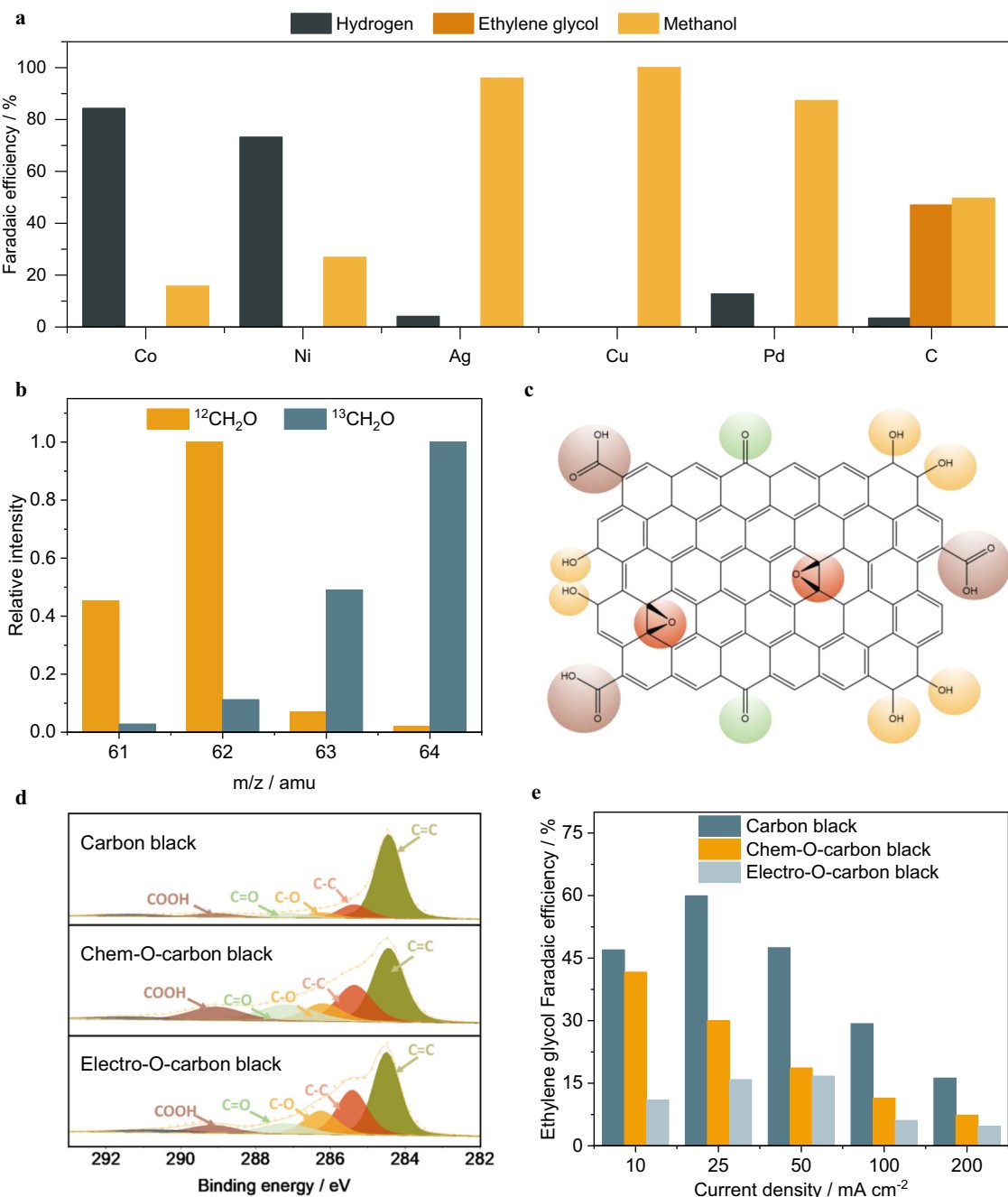

**Fig. 2 | Electrochemical formaldehyde reduction to ethylene glycol on carbon catalysts. a** Selectivity of electrochemical formaldehyde reduction on various catalysts at a current density of 25 mA cm$^{-2}$. The experiment was performed in 37 wt% formaldehyde solution containing 1 M sodium acetate as supporting electrolyte under ambient temperature and pressure. **b** Mass spectra of ethylene glycol produced in formaldehyde reduction on carbon catalysts. The $^{13}$C-labeled formaldehyde ($^{13}$CH$_2$O) and non-labeled formaldehyde ($^{12}$CH$_2$O) were used as starting feedstock, respectively. **c** Schematics of oxygen-containing functional groups on carbon. **d** C 1 s X-ray photoelectron spectroscopy of carbon black and oxygen-doped carbon black, including chemically oxidized carbon black (Chem-O-carbon black) and electrochemically oxidized carbon black (Electro-O-carbon black). **e** Ethylene glycol Faradaic efficiency on carbon black and oxygen-doped carbon black at various current density.

for formaldehyde reduction under identical conditions. As shown in Supplementary Fig. S6, the rise of oxygen-containing functional groups undermines the ethylene glycol formation while promotes the hydrogen evolution reaction, which can be attributed to oxygen-containing functional groups on carbon that enhance the hydrophilicity of carbon material thus promotes the competing hydrogen evolution reaction[20,21]. The blank carbon paper is also tested for formaldehdye electroreduction and exhibits much higher overpotential and higher hydrogen FE (Supplementary Fig. S7). This can be attributed to the high loading of PTFE in carbon paper covering the active sites of carbon catalysts.

To gain a better understanding of ethylene glycol formation, the effects of concentration and temperature on formaldehyde reduction are investigated. Since the commercial formaldehyde solution typically contains 37 wt% formaldehyde, 15 wt% methanol stabilizer and 48 wt% water, the concentration dependence was investigated in 10 wt%, 20 wt%, 30 wt%, and 37 wt% formaldehyde solution containing 1 M sodium acetate as a supporting electrolyte. As shown in Fig. 3a, the

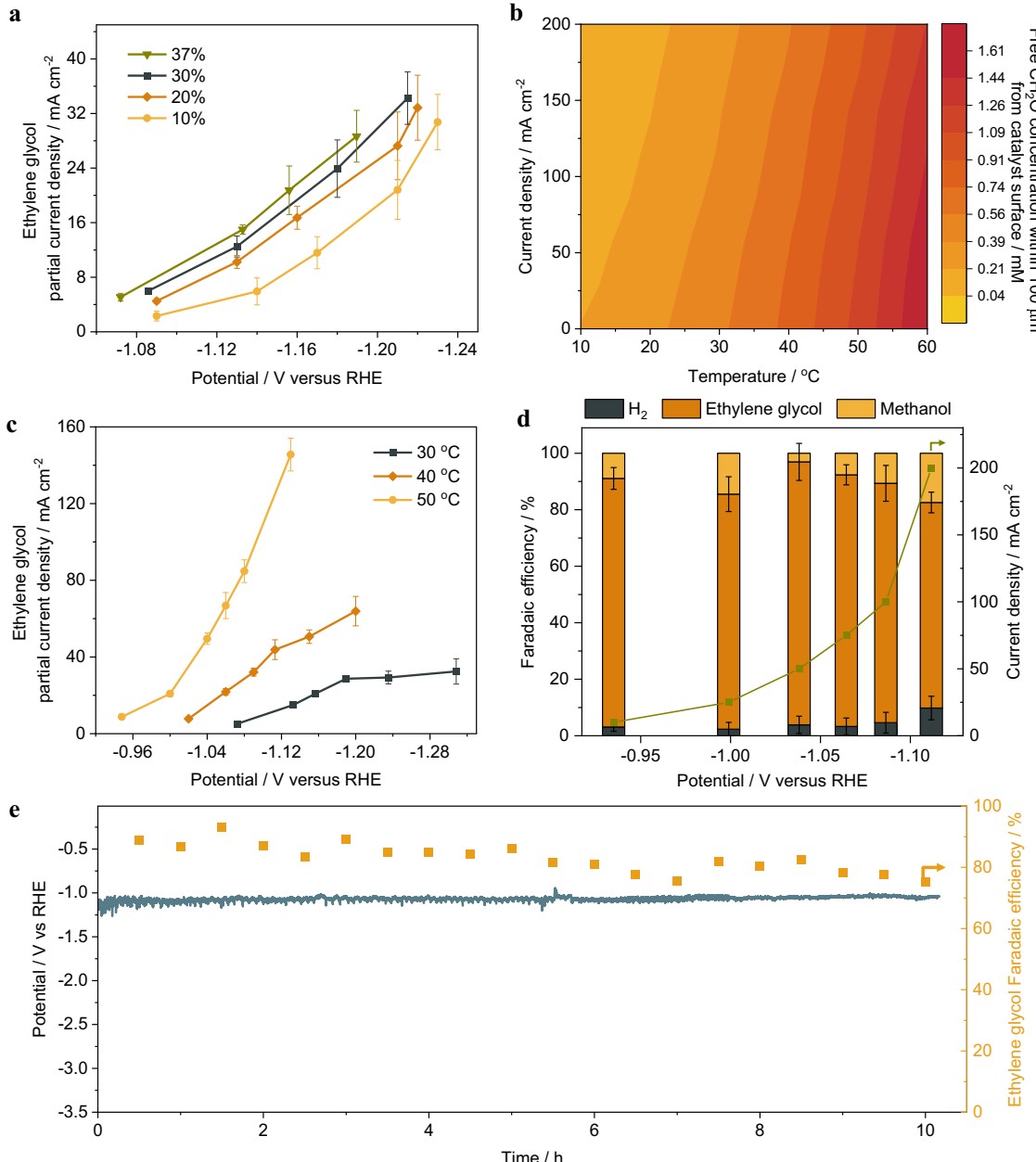

**Fig. 3 | The concentration and temperature dependence of formaldehyde electroreduction. a** Concentration dependence of formaldehyde electroreduction in 10 wt%, 20 wt%, 30 wt% and 37 wt% formaldehyde solution. **b** Calculated free formaldehyde concentrations within 100 μm from the catalyst surface changing with temperatures and current densities. **c** Temperature dependence of formaldehyde reduction at 30 °C, 40 °C, 50 °C, respectively. **d** Performance of formaldehyde electroreduction on carbon under the optimal condition (37% formaldehyde solution containing 1 M sodium acetate as supporting electrolyte, 50 °C). **e** 10-h stability test of formaldehyde electroreduction on carbon at constant current density of 100 mA cm⁻² in 37 wt% formaldehyde solution containing 1 M sodium acetate as supporting electrolyte at 50 °C. Error bars represent the standard deviation in three independent measurements.

partial current density of ethylene glycol increased with the concentration of formaldehyde. A non-exponential increase is evident under all concentrations, suggesting a mass transfer limitation even at a concentration as high as 37 wt%. At high current densities, hydrogen evolution and methanol formation become more dominant reactions. We hypothesize that it is because of low equilibrium concentration of free formaldehyde in aqueous solution due to the reversible hydration reaction of formaldehyde in water.

The presence of formaldehyde in aqueous solutions is a mixture of its hydrated form, methanediol ($CH_2(OH)_2$) and dissolved free formaldehyde: $CH_2O(g) + H_2O(l) \rightleftharpoons CH_2(OH)_2(l)$. To proceed with electroreduction of formaldehyde, methanediol needs to go through the reverse reaction to form free formaldehyde, and then the free

formaldehyde is reduced to ethylene glycol or methanol. We hypothesize that only free formaldehyde in the carbon catalyst layer can be electrocatalytically reduced. Thus, the limiting current density is determined by the concentration of free formaldehyde in the catalyst layer. To estimate the catalyst layer free formaldehyde concentration, we constructed a reaction kinetics model considering the reversible hydration reaction, electrocatalytic reaction in the carbon catalyst layer and mass transfer of the formaldehyde from the bulk electrolyte solution. (See detailed discussion in the Supplementary Information). The modeling results (Fig. 3b) show that at 200 mA cm⁻² the concentrations of free formaldehyde at 50 °C is 5 times higher than that at 20 °C. The majority of formaldehyde is presented as hydration form of methanediol, which is in line with the literature[22]. The concentration of

free formaldehyde increases as the temperature rises because the dehydration of methanediol is an endothermic reaction. Based on the model, we conducted the formaldehyde electroreduction at elevated temperatures and found that the partial current density of ethylene glycol increased from 15.0 mA cm$^{-2}$ to 145.56 mA cm$^{-2}$ at $-1.13$ V vs. RHE when the temperature increased from 30 °C to 50 °C (Fig. 3c, d and Supplementary Fig. S9), consistent with the modeled local free formaldehyde concentration.

The pH dependence of formaldehyde electroreduction is also investigated using buffered solutions with a pH ranging from 2.6 to 6.8. Alkaline conditions favor the Cannizzaro reaction of formaldehyde, leading to the production of formic acid and methanol[23]. Consequently, pH dependence study is performed exclusively in acidic to neutral environments. The concentration of Na$^+$ is maintained at 0.1 M to exclude any potential cation effect. The partial current densities of ethylene glycol production are plotted against the applied potentials on a standard hydrogen electrode (SHE) scale. As shown in Supplementary Fig. S10, the formation of ethylene glycol is pH-independent, suggesting that proton is not involved in the rate-determining step of formaldehyde reduction to ethylene glycol. In contrast, hydrogen and methanol formation exhibit a strong pH dependence and are favored in acidic conditions. The pH dependence study indicates that neutral condition is more favorable to suppressing undesired side reactions in formaldehyde electroreduction.

The formaldehyde electroreduction is conducted in 37% formaldehyde solution containing 1 M sodium acetate as supporting electrolyte at 50 °C, 92% ethylene glycol FE was achieved at $-1.04$ V vs. RHE. Over 70% ethylene glycol FE was observed in a wide potential range, as shown in Fig. 3d. The long-term stability of formaldehyde electroreduction on carbon was examined at a constant current density of 100 mA cm$^{-2}$ in 37 wt% formaldehyde solution containing 1 M sodium acetate as a supporting electrolyte. The results show a stable potential ($-\!-1.07$ V vs. RHE) and ethylene glycol FE (>76%) maintained over a span of 10 h (Fig. 3e). SEM characterization of the post-reaction carbon catalyst confirmed no noticeable morphology change of the catalyst after the stability test (Supplementary Fig. S11). The XPS spectra suggests that there is no metal deposition after long-term stability test except for sodium and potassium that from the electrolyte (Supplementary Fig. S12).

## Methanol partial oxidation to formaldehyde

Producing ethylene glycol from methanol substantially reduces the cost of raw material, representing an economically competitive pathway for ethylene glycol production. Thus, methanol partial oxidation is coupled

with formaldehyde reduction, enabling electrosynthesis of ethylene glycol through a redox reaction. Formaldehyde has been observed as a side product of methanol oxidation in direct methanol fuel cell (DMFC)[24]. Previous study suggests that formaldehyde formation increased with methanol concentrations. Formaldehyde formation on Pt solid catalyst increased five times when the methanol concentration is increased from 0.05 M to 0.3 M, and regular direct methanol fuel cell is usually operated under a methanol concentration below 3 M[25]. Herein, we investigated the methanol partial oxidation in anhydrous methanol to maximize formaldehyde production and 1 M sodium acetate is added as supporting electrolyte. Pt nanoparticles and Pt microparticles were synthesized and examined for formaldehyde electrosynthesis in a two-compartment flow electrolyzer (Supplementary Fig. S13). Previous study reveals that methanol dehydrogenation reaction to formaldehyde is the initial step of methanol oxidation, followed by further dehydrogenation of formaldehyde to CO and CO oxidation to CO$_2$ using water as the oxygen source. CO$_2$ is not observed as a side product, suggesting there is no full oxidation of methanol in this study since there is no water as oxygen source. Formaldehyde is the major product determined by fluorescence spectroscopy using Nash method[26]. In the anhydrous methanol solution, formaldehyde partially exists as methylal (H$_3$COCH$_2$OCH$_3$) as there is an equilibrium reaction between methylal and free formaldehyde: H$_3$COCH$_2$OCH$_3$ + H$_2$O $\leftrightarrow$ HOCH$_2$OCH$_3$ + CH$_3$OH $\leftrightarrow$ CH$_2$O + CH$_3$OH. After the methanol partial oxidation reaction, cation-exchange resin (Amberlyst, 15 hydrogen form) is added to the post-reaction anolyte along with DI water to promote the release of free formaldehyde before quantification (see details in the methods section). Pt nanoparticles exhibits higher current density than Pt microparticles (Fig. 4a). However, Pt microparticles show significantly enhanced selectivity towards formaldehyde. As shown in Fig. 4b, the maximum formaldehyde Faradaic efficiency on Pt microparticles is 75%, which is 2.5 times as high as that on Pt nanoparticles. The platinum mesh also shows activity in the partial oxidation of methanol to formaldehyde (Supplementary Fig. S14), with a Faradaic efficiency of up to 33% for formaldehyde production. However, a high potential exceeding 3 V versus RHE is required to attain a current density of 5 mA cm$^{-2}$. As shown in Supplementary Fig. S15, X-ray diffraction (XRD) pattern reveals that both Pt nanoparticles and microparticles exhibit similar crystal structure and all the diffraction peaks align well with metallic Pt standard (PDF#04-0802). The enhanced formaldehyde selectivity can be attributed to the particles size difference as large size particles expose more basal plane sites while fine nanoparticles are dominated by edge sites. Edge sites of Pt have intrinsic high activity toward methanol fully oxidation while

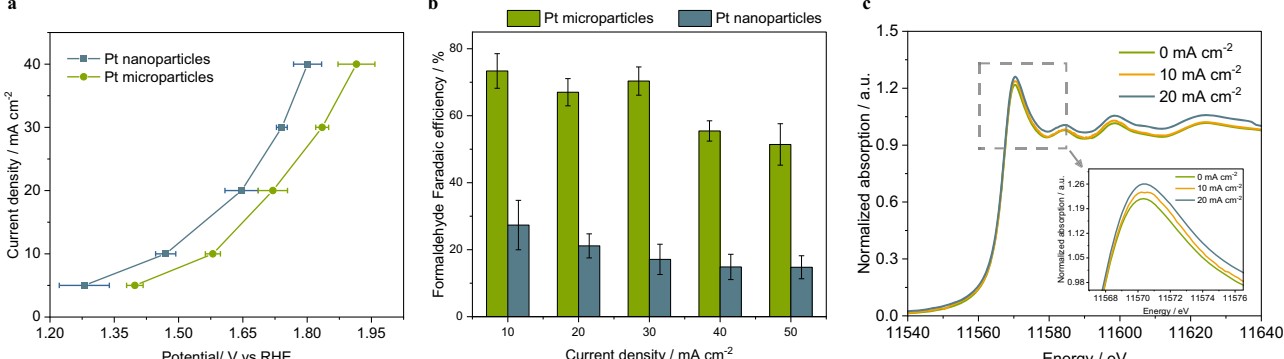

**Fig. 4 | Methanol electrochemical partial oxidation to formaldehyde. a** Total current density versus applied potential for methanol partial oxidation reaction on Pt nanoparticles and Pt microparticles, respectively. Anhydrous methanol containing 1 M sodium acetate is used as supporting electrolyte. **b** Formaldehyde Faradaic efficiency for methanol partial oxidation reaction on Pt microparticles and

Pt nanoparticles. **c** In-situ X-ray absorption near edge structure (XANES) spectra of Pt microparticles under the methanol electrochemical partial oxidation reaction condition in anhydrous methanol containing 1 M sodium acetate. Error bars represent the standard deviation in three independent measurements.

basal plane sites have relative lower formaldehyde binding energy. Thus, formaldehyde is easily desorbed as the final product from Pt with large particle size[22,27]. In-situ X-ray absorption spectroscopy is applied to investigate the oxidation state of platinum in real-time during the methanol oxidation reaction conducted in anhydrous methanol containing 1 M sodium acetate as supporting electrolyte. A constant current density of 10 mA cm$^{-2}$ and 20 mA cm$^{-2}$ was applied, respectively. The in-situ X-ray absorption near edge structure (XANES) spectra of Pt catalysts was compared with Pt(0) and Pt(II) standard (Supplementary Fig. S16), suggesting that Pt mainly remains the metallic state under the methanol partial oxidation reaction. Notably, an increase in the intensity of the white-line in the Pt L3-edge spectra is observed with rising current density (Fig. 4c), suggesting that platinum exhibits a more oxidized characteristic under higher potentials[28]. Our proof-of-concept study employed Pt catalysts for methanol partial oxidation to generate formaldehyde. Future research will focus on exploring the use of Platinum Group Metal (PGM)-free catalysts or incorporating secondary metals, with the objective of reducing Pt usage.

## Electrosynthesis of ethylene glycol from methanol in a membrane-electrode-assembly based electrolyzer

The feasibility of ethylene glycol electrosynthesis directly from methanol is investigated by coupling anodic methanol partial oxidation with formaldehyde electroreduction in a single electrolyzer. Although both methanol partial oxidation and formaldehyde electroreduction show a reasonable half-cell potential in the three-electrode configuration after iR compensation, the full cell voltage is extremely high due to the high internal resistance caused by poor solubility and conductivity of supporting electrolyte in organic phase. The conductivity of 1 molar sodium acetate in 37 wt% formaldehyde solution is 14.27 μS cm$^{-1}$, approximately 5 times lower than that in water. In a standard two-compartment flow electrolyzer configuration, even a thin layer of organic electrolyte of ~ 2 mm between catalyst and membrane accounts for significant ohmic loss, resulting in a full cell voltage of ~7.2 V at 100 mA cm$^{-2}$. By using a membrane electrode assembly-based electrolyzer, as shown in Fig. 5a, catalyst layer is hot-pressed onto the ionic conductive membrane and the internal resistance is lowered to 0.694 Ω, compared with 5.392 Ω in two-compartment flow electrolyzer. As a result, the full cell voltage in MEA-based electrolyzer is reduced to 3.2 V at 100 mA cm$^{-2}$ (Fig. 5b). The techno-economic assessment suggests that operating cell voltage plays a significant role in total cost of ethylene glycol production. Reducing the cell voltage to 3.2 V enables ethylene glycol electrosynthesis from formaldehyde has a competitive price compared with conventional manufacturing route (Fig. 5c). In membrane electrode assembly based electrolyzer, a maximum Faradaic efficiency of 75.6% and 73.4% for ethylene glycol and formaldehyde are achieved in methanol partial oxidation reaction and formaldehyde reduction reaction, respectively (Fig. 5d). It is noticed that formaldehyde production via methanol partial oxidation prefers low current densities (i.e., low overpotentials), whereas ethylene glycol production via formaldehyde electroreduction reaches a peak formaldehyde Faradaic efficiency of 75% at a current density of 100 mA cm$^{-2}$, where the corresponding formaldehyde Faradaic efficiency is about 50% (about 25% lower than the maximum formaldehyde Faradaic efficiency in methanol partial oxidation). Those findings could guide future design of

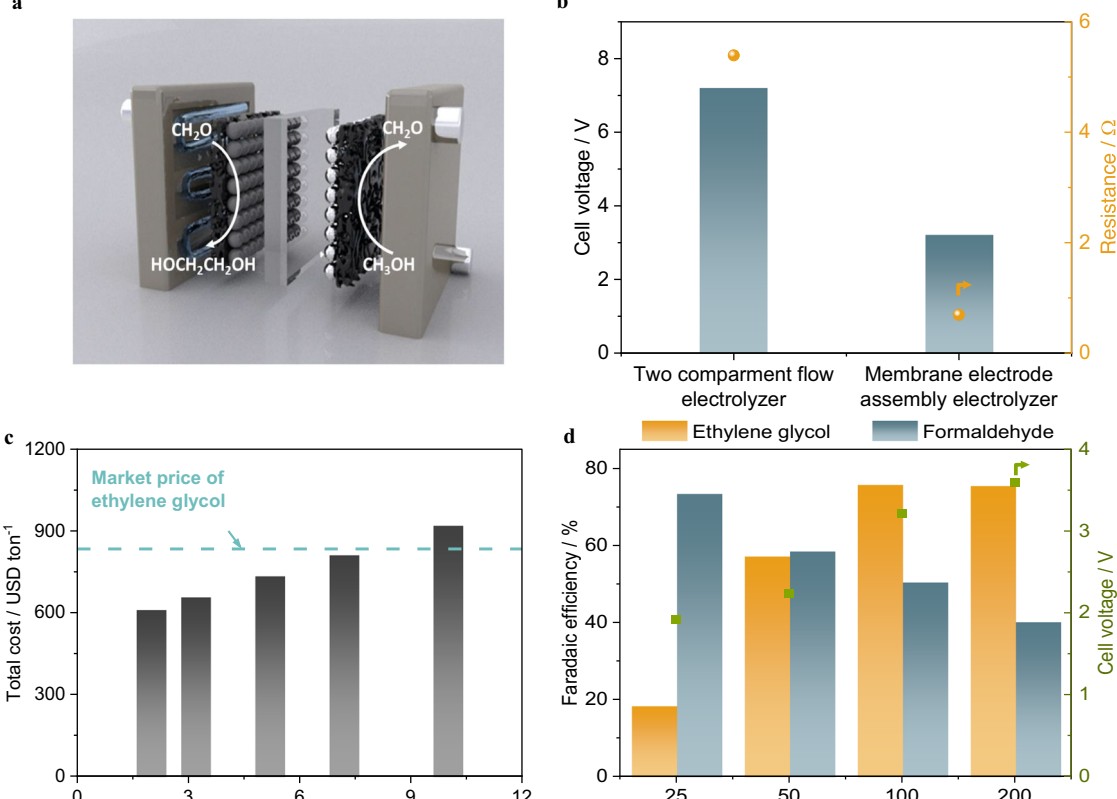

**Fig. 5 | Electrosynthesis of ethylene glycol in a single membrane-electrode-assembly (MEA) electrolyzer. a** Schematics of coupling methanol partial oxidation and formaldehyde electroreduction in a single membrane electrode assembly electrolyzer. **b** Cell voltage and internal resistance of formaldehyde electroreduction reaction in different cell configurations. **c** Dependence of total cost of ethylene glycol production from methanol on operating cell voltage. **d** Performance of ethylene glycol electrosynthesis via coupling anodic methanol partial oxidation with formaldehyde electroreduction in a single membrane-electrode-assembly electrolyzer.

paired electrolysis processes for ethylene glycol synthesis from methanol. This process can be integrated with the emerging $CO_2$ hydrogenation-based methanol production to improve the overall sustainability, showcasing a sustainable route from $CO_2$ to ethylene glycol.

## Reaction mechanism of C-C coupling

The C-C coupling step plays a crucial role in synthesizing ethylene glycol from the C1 product, however, there is little study on the mechanism of C-C coupling of formaldehyde. There are two potential pathways for C-C coupling in formaldehyde electroreduction, as shown in Fig. 6a. In the first case, C-C coupling happens through a formose reaction, where a condensation reaction of two formaldehyde molecules forms glycolaldehyde, followed by glycolaldehyde hydrogenation to ethylene glycol[29,30]. The other pathway is C-C coupling occurs after an initial proton transfer step, where two $CH_2OH$ intermediates react to form ethylene glycol. Formose reaction is observed in prebiotic synthesis in nature to make glycolaldehyde and higher carbohydrates from formaldehyde, catalyzed by divalent metal cations such as $Ca^{2+}$ and $Mg^{2+}$ (see the mechanism in Supplementary Fig. S17)[14]. Because the glycolaldehyde formation has sluggish kinetics and is only reported to be promoted by the UV-radiation and free radical[15–17], we hypothesize that it is not the key intermediate in formaldehyde electroreduction to ethylene glycol. To rule out the potential involvement of glycolaldehyde, we conducted a detailed analysis using operando flow electrolyzer mass spectroscopy (FEMS, as shown in Supplementary Fig. S18) to probe reaction intermediates from approximately $100\,\mu m$ from the electrode surface during the electrocatalytic reaction. Current densities at $10\,mA\,cm^{-2}$, $25\,mA\,cm^{-2}$, $50\,mA\,cm^{-2}$, and $100\,mA\,cm^{-2}$ were

applied, and only a trace amount of glycolaldehyde was detected under formaldehyde electroreduction reaction (Fig. 6b).

In the $^{13}C$-labeling experiments, formaldehyde electroreduction was performed using 4 wt% $^{12}CH_2O$ and $^{13}CH_2O$ in 1 M sodium acetate (supporting electrolyte), respectively. Then, 0.1 M non-labeled glycolaldehyde is added to $^{13}CH_2O$. All experiments were performed at a constant current density of $10\,mA\,cm^{-2}$ for 10 hours, and the resulting ethylene glycol in the electrolyte was analyzed by GC-MS. As shown in Fig. 6c, when $^{12}C$-glycolaldehyde is added into the $^{13}CH_2O$, the mass fragmentation ratio of ethylene glycol remains identical to the signal we observed in the $^{13}CH_2O$ case, suggesting that glycolaldehyde is not involved in the formation of ethylene glycol. A trace amount of glycolaldehyde observed using operando flow electrolyzer mass spectroscopy is likely due to the formose reaction that naturally occurs slowly in the formaldehyde solution. The $^{13}C$-labeling experiments provide clear insights that glycolaldehyde is not the reaction intermediate towards ethylene glycol formation.

Since methanol is also a product of formaldehyde reduction, $^{13}C$-labeled methanol is added to $^{12}CH_2O$ to investigate whether the presence of methanol interferes with the formation of ethylene glycol in formaldehyde electroreduction. The MS spectra showed that both carbons in ethylene glycol were $^{12}C$, indicating that both carbons in ethylene glycol originated from $^{12}CH_2O$ and $^{13}C$-methanol did not participate in ethylene glycol formation. Based on the experimental results, we propose that ethylene glycol formation in formaldehyde electroreduction goes through two steps: a formaldehyde protonation step to form $CH_2OH$, and a C-C coupling step to form ethylene glycol.

To verify the proposed mechanism, we conducted DFT calculations with a focus on the C-C coupling step because it is likely the key

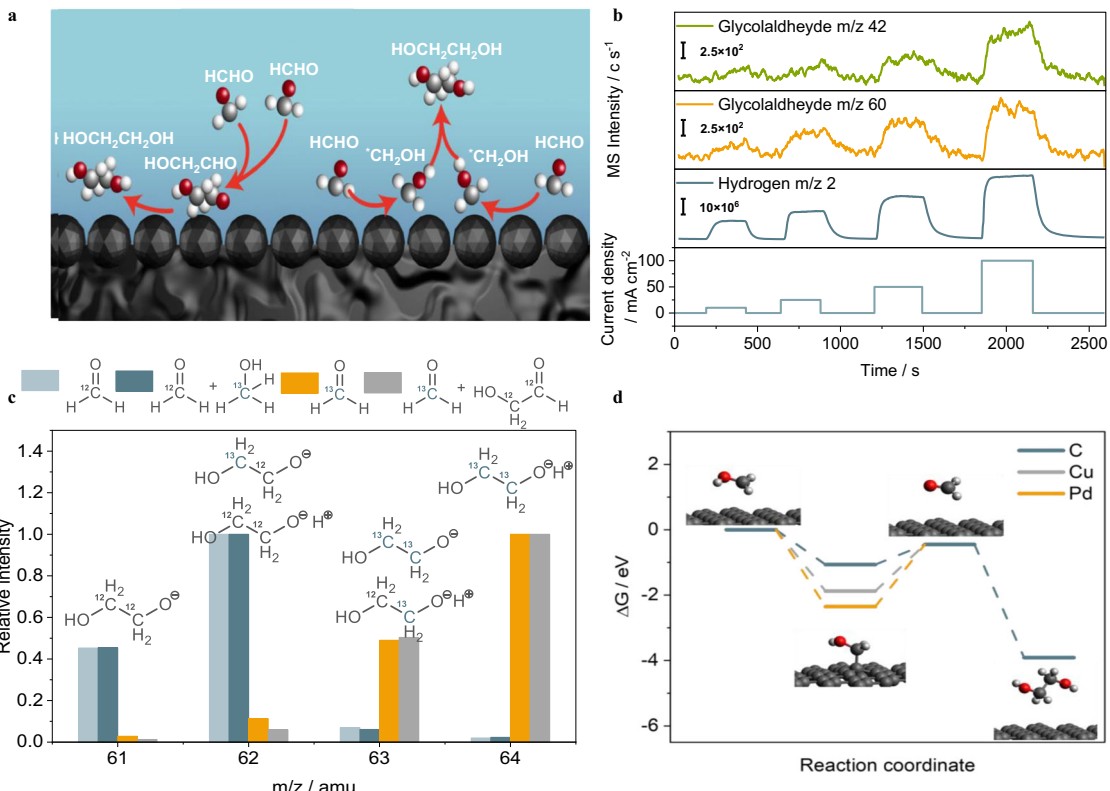

**Fig. 6 | Reaction mechanism study of formaldehyde electroreduction.**
**a** Potential reaction mechanism of ethylene glycol formation through glycolaldehyde and $^*CH_2OH$ intermediate, respectively. **b** Product distribution analysis using operando flow electrolyzer mass spectroscopy of formaldehyde electrochemical reduction on carbon at a constant current density of $10\,mA\,cm^{-2}$, $25\,mA\,cm^{-2}$,

$50\,mA\,cm^{-2}$, and $100\,mA\,cm^{-2}$. **c** Mass spectra of ethylene glycol produced in formaldehyde reduction using $^{12}CH_2O$, $^{13}CH_2O$, $^{13}CH_2O + HO^{12}CH_2^{12}CHO$ and $^{12}CH_2O + ^{13}CH_3OH$ as starting feedstock. **d** Reaction coordinates of formaldehyde electroreduction to ethylene glycol on carbon, copper, and palladium catalysts.

defining the selectivity for ethylene glycol formation. The C-C coupling may occur between two $CH_2OH$, which either remain absorbed on the catalyst or desorbed into the solution before the coupling. We find the direct coupling between two adsorbed $*CH_2OH$ has prohibitively high activation energy on both graphene basal plane and edge site (see supporting information for details). In contrast, coupling between two desorbed $CH_2OH$ is easy, with a barrier of only 0.28 eV. Importantly, the desorption of $CH_2OH$ from the graphene basal plane requires 0.62 eV, which is feasible at room temperature. These results suggest the important role of $CH_2OH$ desorption in forming ethylene glycol. Therefore, we compare the $CH_2OH$ desorption energy between C, Cu, and Pd to understand why carbon is the best catalyst (the most favorable adsorption sites and configurations are presented in Supplementary Fig. S20). As shown in Fig. 6d, the desorption energies of $CH_2OH$ on the (111) surface of Cu and Pd are 1.43 eV and 1.90 eV, respectively, which are much higher than that on the carbon surface (0.62 eV). The stronger adsorption thus impedes the pathway toward the ethylene glycol and steers the reaction towards further hydrogenation of $CH_2OH$ intermediate, leading to methanol formation. The low desorption energy barrier explains why carbon is unique in favoring ethylene glycol formation, while copper and palladium only show selectivity towards methanol.

In this work, we presented an electrochemical route for ethylene glycol synthesis from methanol. The redox reaction is divided into an oxidation reaction converting methanol to formaldehyde, followed by a reduction reaction transforming formaldehyde into ethylene glycol. Carbon-based catalysts are employed at the cathode to reduce formaldehyde into ethylene glycol exhibiting a high selectivity (>90 % FE), while platinum-based catalysts are utilized at the anode to partially oxidize methanol, converting it into formaldehyde with a maximum formaldehyde FE of 75%. The feasibility of ethylene glycol electrosynthesis from methanol is demonstrated by coupling methanol anodic partial oxidation with formaldehyde electroreduction in a membrane electrode assembly-based electrolyzer, which substantially reduced the cell voltage and enhance the energetic efficiency at a current density up to 200 mA cm$^{-2}$. The reaction mechanism of C-C coupling to form ethylene glycol was further investigated through operando flow electrolyzer mass spectroscopy, isotopic labeling experiments and DFT calculations. The results suggest that the desorption of $CH_2OH$ intermediate was identified as the key step leading to ethylene glycol formation,

## Methods

### Electrode preparation

Commercially available carbon black (Vulcan XC 72 R, ~50 nm, Fuel cell store) is used as the catalyst for formaldehyde electroreduction. To remove any potential contaminant on carbon catalysts, Vulcan carbon is first treated with 5 wt% HCl under sonication for 30 min. Then the Vulcan carbon is filtrated and washed with DI water until the pH is 7. After drying in a vacuum oven overnight, the as-treated Vulcan carbon is annealed at 500 °C for 3 h under a 5%H$_2$/Ar atmosphere. In the typical procedure to make an electrode, 25 mg of catalysts are dispersed in a mixture of isopropanol and water (1:1 ratio) with 5 wt% Nafion ionomer dispersion (D1021 Nafion Dispersion, Fuel Cell Store) as the binder. The catalyst ink is sonicated for 30 min prior to use and then dropcasted onto a porous carbon paper (Sigracet 39BB, Fuel cell store). Ti felt is used as substrate for methanol partial oxidation due to its high stability under oxidative potential. The loading is controlled at 0.5 mg cm$^{-2}$. Then the as-prepared electrode is dried at 70 °C overnight to evaporate all the solvent.

Catalysts screening was performed using commercial metal nanoparticles. Co (200 nm, 99.95%), Ni (40 nm, 99.9%), Ag (20 nm, 99.99%) were purchased from US Research Nanomaterials, Inc. Cu nanoparticles (25 nm, 99.99%) and Pd nanoparticles (25 nm, 99.6%) were purchased from Sigma Aldrich. The metal catalysts are loaded

onto titanium fiber felt (Fuel cell store) and the loading was controlled at 0.5 mg cm$^{-2}$ based on the metal mass.

To synthesize the platinum microparticles, platinum chloride is first dissolved in 10 mL DI water and stirred for 20 min, followed by adding sodium tetrahydroborate solution (0.2 g NaBH$_4$ dissolved in 2 mL water) drop by drop[31]. The Pt precursor is slowly reduced by hydrogen released from sodium tetrahydroborate without external stirring. The resulting Pt particles are collected via centrifugation after 3-hour settling and washed with DI water for 3 times. Pt nanoparticles are synthesized using poly-N-vinylpyrrolidone (PVP) as a capping agent[32]. Typically, 50 mL ethylene glycol solution is heated to 120 °C under vigorous stirring. The pH value is adjusted to >13 with 1 M sodium hydroxide. A mixture of hexachloroplatinic acid (5.3 mM) and PVP (91 mM) is slowly added to the ethylene glycol solution. The resulting Pt particles are collected by centrifugation after adding acetone and washed with water three times.

### Formaldehyde electroreduction

The formaldehyde electroreduction is initially performed in a two-compartment microfluidic flow electrolyzer. Two stainless end-plates were used as current collector[33]. The flow channel is 2 mm thick, and the active area of electrode is 1 cm$^2$. Nafion 212 (Fuel cell store) is used to separate the cathodic and the anodic chamber due to its excellent stability in organic solvents. In a typical formaldehyde electroreduction reaction, 37 wt% formaldehyde solution containing 1 M sodium acetate is used as catholyte, and 1 M sodium acetate is fed to the anode. Sodium acetate is chosen as the supporting electrolyte because of its high solubility in formaldehyde solution compared with other supporting electrolytes (i.e., sodium perchlorate and sodium sulfate). Commercial formaldehyde solution (37 wt% formaldehyde with 15% methanol as a stabilizer) is purchased from Sigma-Aldrich. Methanol-free formaldehyde solution is made from paraformaldehyde pyrolysis. To make a methanol-free formaldehyde solution, DI water is heated in a round bottle flask to 60 °C under Ar atmosphere, and paraformaldehyde is added while stirring. The paraformaldehyde suspension is stirred for 10 min, and then 4 mL 1 M sodium hydroxide is slowly added to the solution. The suspension is heated for another 10 min until it turns transparent. Phosphoric acid/acetic acid is added to adjust the pH to neutral. The catholyte and anolyte flow rates are controlled at 0.6 mL min$^{-1}$ using a peristaltic pump (Cole-Parmer). The potential is applied using a potentiostat (Metrohm Autolab PG128N), and the half-cell potential is measured against an external Ag/AgCl reference electrode (Pine research). The Ag/AgCl reference electrode was calibrated against a new Ag/AgCl electrode before and after each experiment using a voltage meter. No significant potential changes were observed during short-term use for formaldehyde electroreduction. The electrolyzer is heated with a heating pad affixed to the stainless-steel endplate. A thermocouple is attached to the electrolyzer to control the temperature. Both catholyte and anolyte are heated in a temperature-controlled oil bath. The temperature is set higher than the electrolyzer to compensate for the heat loss between the electrolyte reservoir and the electrolyzer. The temperature is adjusted according to the environmental temperature to keep the electrolyte feeding into the cell under the exact temperature of the electrolyzer. All the electrolyzer and tubing are insulated using glass wool. The gas product is separated from the catholyte using a gas-liquid separator and fed to an inline gas chromatogram (Multiple Gas Analyzer #5, SRI Instruments) equipped with a HayeSep D column. Hydrogen is quantified using a thermal conductivity detector (TCD). The liquid product is analyzed using $^1$H NMR (Bruker AVIII 600 MHz) via a pre-saturated water suppression method. To prepare the NMR sample, 0.5 mL liquid sample is mixed with 0.1 mL D$_2$O containing 500 ppm DMSO as an internal standard.

## Methanol partial oxidation

Methanol partial oxidation is first conducted in a two-compartment flow electrolyzer in anhydrous methanol with 1 M sodium acetate as the supporting electrolyte. Pt/C and Pt nanoparticles/microparticles are used as cathode and anode. Thus, methanol partial oxidation reaction is coupled with hydrogen evolution reaction. The flow rates are controlled by peristaltic pumps. The electrolyte and electrolyzer temperatures are set to 50 °C to match the optimal condition for formaldehyde electroreduction. An external Ag/AgCl electrode is used as reference electrode. The liquid product is collected directly from the anolyte. In anhydrous methanol solution, formaldehyde partially exists as methylal ($H_3COCH_2OCH_3$). And there is an equilibrium reaction between methylal and free formaldehyde: $H_3COCH_2OCH_3 + H_2O \leftrightarrow HOCH_2OCH_3 + CH_3OH \leftrightarrow CH_2O + CH_3OH$. To shift the reaction towards methylal hydrolysis to release free formaldehyde, DI water is added to the liquid sample with a 5:1 mass ratio after the methanol partial oxidation reaction[34], followed by adding 10 wt% ion-exchange resin (Amberlyst® 15 hydrogen form, Sigma Aldrich) to catalyze the hydrolysis process at 50 °C for 3 h[35]. Formaldehyde is quantified by the Nash method using UV-Vis Spectrophotometer (Cary 60, Agilent)[26,36]. In a typical Nash method, a 1 mL sample is mixed with 2 mL of Nash solution composed of 2.0 M ammonium acetate, 0.05 M acetic acid, and 0.02 M acetylacetone. Formaldehyde is first converted by acetylacetone to 3,5-diacetyl-1,4-dihydrolutidin (DDL) in the presence of an ammonium acetate buffered solution through the Hantzsch reaction. The resulting DDL is a fluorescence-active species that can be detected by fluorescence spectroscopy. The Hantzsch reaction is accelerated by heating in an oil bath at 100 °C for 10 min and left at room temperature to react overnight. The sample is analyzed by UV-vis spectroscopy. The maximum absorption for formaldehyde is at a wavelength of 411 nm. The liquid sample is diluted multiple times before measurements.

## $^{13}$C-isotope labeling experiment

The $^{13}$C-formaldehyde ($^{13}CH_2O$, 20 wt% in $H_2O$, 99 atom% $^{13}$C) is purchased from Sigma-Aldrich. Due to the scarcity of $^{13}$C-labeled formaldehyde, it is only available in 1 mL package with a concentration of 20 wt%. The flow-type experiment in this study requires >20 mL electrolyte at a time. Thus, the isotope labeling experiment was carried out in a batch cell with a fixed volume of 5 mL. The electrolyte contained 4 wt% $^{13}CH_2O$ in water diluted from 1 mL 20 wt% $^{13}$C-formaldehyde. 1 M sodium acetate was added as supporting electrolyte and 5 mL of 1 M sodium acetate aqueous solution was used as an anolyte. Vulcan carbon on porous carbon paper with an active area of 1 cm$^2$ is used as the electrode for the cathode and $IrO_2$ on Ti felt is used as the anode. A constant current of 10 mA cm$^{-2}$ is applied for 10 h. After the electrolysis, a rotary evaporator with a dry ice condenser was used to evaporate all the water from the resulting electrolyte. The water bath temperature is set to 40 °C, and the liquid sample is rotavaped for 15 min to remove all the water. 2 mL acetonitrile was added to extract ethylene glycol. The as-prepared liquid sample is analyzed by GC-MS (Agilent 59771 A) focusing on the mass fragment patterns of resulting ethylene glycol.

## DFT calculations

DFT calculations were conducted with the Vienna ab initio simulation package (VASP)[37], in which the implicit solvation was also implemented through the VASPsol patch[38,39]. Perdew-urke-Ernzerhof (PBE) functional with D3 van der Waals correction was employed with 520 eV cutoff energy of plane wave[40,41]. To simulate the catalyst-electrolyte interface, we use the constant-potential model (CPM)[42] where the electron number is adjusted to achieve a potential of −1.5 V vs SHE at which there is optimal productivity. The barrier for C-C coupling between two adsorbed $CH_2OH$ is obtained using slow-growth approach while that in solution is calculated using nudged elastic band (NEB) method[43–45].

## Flow electrolyzer mass spectrometry (FEMS)

The operando flow electrolyzer mass spectrum was utilized to identify the gaseous and volatile reaction intermediate/products. A PEEK capillary (McMaster, inner diameter 0.25 mm) was covered by a hydrophobic PTFE membrane (TISCH, pore size 200 μm) to separate the aqueous electrolyte from the target volatile and gaseous products that enter the vacuum chamber. The capillary probe is placed at ~100 μm from the electrode surface in a three-compartment flow electrolyzer, enabling real-time detection of gaseous and volatile products under the reaction condition[46]. The detailed schematic of the system is shown in Supplementary Fig. S18. Differential pump applies DUO 20 M (Pfeiffer) and mass spectrometer utilized Hidden Quadruple. A secondary electron detection voltage of 1700 V with and emission current of 200 μA was utilized to generate an ionization potential of 70 eV to ionize the product samples.

## Techno-economic analysis

The techno-economic analysis is conducted based on a previous published model[47]. The parameters used in this model are listed in Supplementary Table S2. The experimental performances observed in this study served as the basis for determining the performance parameters, which included current density, cell voltage, and Faradaic efficiency. To determine the production cost, a selling price was evaluated such that the net present value at the end of the process life was zero, assuming a plant lifespan of 20 years. A production scale of 10,000 kg per day was selected to maintain consistency with previous work and to reflect a commercially viable process. The cost analysis shown in Fig. 4b was conducted using a 20-year plant lifespan and factoring in both capital and operating costs.

## Data availability

All data supporting the findings of this study are available within the article, as well as the Supplementary Information file. All other data supporting the findings of the study are available from the corresponding authors upon reasonable request.

## Code availability

All codes are available from the authors upon reasonable request.

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

## Acknowledgements

The authors at the University of Delaware would like to acknowledge the financial support from Delaware Energy Institute. X.M. would like to thank the support from National Key Research and Development Program of China (2018YFA0704501). R.X. thanks Kentaro Hansen for helping with hot-pressing. This research used resources at the 8-ID Beamline of the National Synchrotron Light Source II, a US Department of Energy Office of Science User Facility operated by Brookhaven National Laboratory under contract no. DE-SC0012704. R.W. and Y.L. acknowledge the support by NSF (1900039, 2029442), ACS PRF (60934-DNI6), and the Welch Foundation (F-1959-20210327). The calculations used computational resources at ACCESS and TACC.

## Author contributions

R.X. conceived the idea and designed the experiments. R.W. performed the DFT calculations. R.X. and B.H. conducted the operando flow electrolyzer mass spectroscopy experiments. A.L. performed the SEM

characterizations. R.X. performed the data analysis and prepared the first draft. F.J. revised the manuscript. Y.L. supervised computational modeling efforts. F.J. and X.M. supervised the whole project. All authors commented on the final version of the manuscript.

## Competing interests

The authors declare no competing interests.
