## [Peer Review File · Nature Communications]

REVIEWER COMMENTS

Reviewer #1 (Remarks to the Author):

The present manuscript reported the oxidation of methanol to formaldehyde and formaldehyde to glycol. This is a significant experimental development, demonstrating the excellent performance of both reactions with two simple catalysts. Overall, the manuscript is interesting, yet the content needs to be improved to be fully convincing: In my opinion the manuscript can be published after major revisions on the following points which need clarification/discussion.

- 1、 According to the manuscript, the metal particles and carbon black are respectively loaded on two kinds of carriers. Please explain the reason. Besides, the related repeated experiments and error bars are needed.
- 2、 Why the pH was ranging from 2.6 to 6.8 but not from acid to alkaline in the dependence of formaldehyde electroreduction?
- 3、 Please explain why temperature can seriously affect the current of formaldehyde to produce glycol. And please give the Faraday efficiency of other temperatures and the relationship between current density and potential in the supporting literature.
- 4、 Can similar effects be achieved if platinum plates are used to replace catalysts in the experiment of methanol preparation of formaldehyde? It is suggested that the performance of platinum sheet should be supplemented. As for Figure 4c, please give the marking and local magnification of pt, otherwise it cannot be convinced that pt is in metallic state.
- 5、 The effect of the membrane electrode experiment is not well. Can we change to series system or use H-cell for two reactions at the same time?

Reviewer #2 (Remarks to the Author):

In the manuscript entitled: " Electrosynthesis of Ethylene Glycol from C1 Feedstocks in a Flow Electrolyzer" the authors has reported an electrochemical approach for the selective synthesis of ethylene glycol from methanol. The topic has been conveniently introduced, and it is very relevant not only academically, but also industrially as well as timely. The obtained conclusions have been adequately supported by experimental and theoretical results. Advanced characterization techniques, including in situ and in operando spectroscopic techniques, have been used to disclose the reaction mechanism, which results have been conveniently supported by theoretical calculations. Overall, those are very interesting and notable results, the research is properly driven

and the manuscript is well written and easy to follow. However, there is a main issue related with the nature of the actual active sites in both electrodes that, in my opinion, remains unclear.

In one hand, the authors have demonstrated that after HCL and thermal treatment the carbon black catalyst still has residual O functional groups. Which is the actual O content? have the authors quantify? Moreover, the author has prepared two different catalysts with a higher O content, prepared by two different routes, and exhibiting different oxygenated functional groups. These catalysts have resulted in lower FE or performance than the original one, ruling out the O functional groups as active sites. However, it is still unclear which are active sites. From theoretical calculations, the graphenic basal planes seems to be more active than edges. Then, can the authors further demonstrate this experimentally using pure, ideal graphene or, alternatively, with a further reduced carbon black?? So, instead of oxidizing, as they have already done, trying to remove as much oxygenated moieties as possible.

On the other hand, it seems clear that Pt planes are more active for MeOH oxidation than the edges. But, can the authors comment on the activity of the different facets?? is there any preferential orientation in the Pt microparticles?? it can be elucidated which could be the most active facet experimentally using single crystals or theoretically?

In conclusion, I recommend the publication of the report in this journal after careful revision of these issues.

Reviewer #3 (Remarks to the Author):

The manuscript describes a very interesting alternative for the synthesis of ethylene glycol, a very important chemical in the chemical industry and responsible for the significant production of CO₂ emissions. The authors suggest synthesizing ethylene glycol in from methanol via formaldehyde. This process will happen in two steps: methanol incomplete oxidation to formaldehyde on a Pt-based electrode and formaldehyde reduction to ethylene glycol on a carbon-based catalyst. In addition to the novelty and scientific relevance, the manuscript is extremely complete. The authors investigate the reaction mechanism by using labelled compounds, DFT and operando analytics in addition to the optimized reaction conditions (pH, temperature and concentration).

My main concern with the manuscript, being a communication, is the amount of information in the main text. For example, the investigation of the active site of carbon-based material can be shortened in the main manuscript and moved to the SI. Moreover, I would suggest presenting the results for the Reaction mechanism of C-C coupling together with the Formaldehyde

electroreduction to ethylene glycol in a summarized manner. Strategies like this will improve the readability and understanding of the paper.

In addition, I would like the authors to address the following general points:

1- Can the authors comment on the impact of the origin of methanol on the overall sustainability of the process?

2- Can the authors comment on the long-term stability of the electrolyzer? I that formaldehyde transport through the existent comments membranes and the use of organic solvents will not be optimal for the stability of the electrolysis.

3- The authors should provide further information on the external reference electrode used and how they overcome the issues of the organic solvents.

4- Can the authors comment on the use of Pt-based electrodes, when all the scientific community tries to avoid the use of PGM?

POINT-BY-POINT RESPONSE TO REVIEWER COMMENTS

Reviewer #1 (Remarks to the Author):

The present manuscript reported the oxidation of methanol to formaldehyde and formaldehyde to glycol. This is a significant experimental development, demonstrating the excellent performance of both reactions with two simple catalysts. Overall, the manuscript is interesting, yet the content needs to be improved to be fully convincing: In my opinion the manuscript can be published after major revisions on the following points which need clarification/discussion.

Reply: We thank the reviewer for supporting this work and below is our response to the comments.

1、 According to the manuscript, the metal particles and carbon black are respectively loaded on two kinds of carriers. Please explain the reason. Besides, the related repeated experiments and error bars are needed.

Reply: The selection of substrates for the cathode and anode catalysts in our study, namely carbon paper and titanium (Ti) felt respectively, is informed by their inherent characteristics and their suitability for different reactions. When carbon paper is used as substrate for anodic methanol oxidation reaction, carbon paper oxidation is observed at a current density of $>200 \text{ mA cm}^{-2}$. In contrast, Ti felt, which is commonly used for anodic reactions like oxygen evolution in water electrolyzer, demonstrates notable stability under oxidative potential. As such, carbon paper and Ti felt are chosen for formaldehyde reduction and methanol oxidation respectively due to their optimal performance under the respective reaction conditions. The error bars representing standard deviation from three independent experiments are added to the main text.

The following context is inserted into the updated manuscript to clarify rationale of substrate selection.

“Ti felt is used as substrate for methanol partial oxidation due to its high stability under oxidative potential.”

2、 Why the pH was ranging from 2.6 to 6.8 but not from acid to alkaline in the dependence of formaldehyde electroreduction?

Reply: Alkaline environments facilitate the Cannizzaro reaction of formaldehyde, which leads to the generation of formic acid and methanol.¹ Consequently, the pH investigation is exclusively conducted within acidic to neutral conditions to avoid this side reaction.

The following context is inserted into the updated manuscript to clarify rationale of pH range selection.

“Alkaline condition favors the Cannizzaro reaction of formaldehyde, leading to the production of formic acid and methanol.¹ Consequently, pH dependence study is performed exclusively in acidic to neutral environments.”

Reference:

1. Peng XD, Barteau MA. Adsorption of formaldehyde on model magnesia surfaces: evidence for the Cannizzaro reaction. *Langmuir* 1989, 5(4): 1051-1056.

3、 Please explain why temperature can seriously affect the current of formaldehyde to produce glycol.

And please give the Faraday efficiency of other temperatures and the relationship between current density and potential in the supporting literature.

Reply: Formaldehyde, when dissolved in water, predominantly exists as its hydrated form, methanediol ($\text{CH}_2(\text{OH})_2$), as represented by the equilibrium reaction: $\text{CH}_2\text{O}(\text{g}) + \text{H}_2\text{O}(\text{l}) \rightleftharpoons \text{CH}_2(\text{OH})_2(\text{l})$. The electroreduction of formaldehyde necessitates methanediol to revert to its formaldehyde form, which then undergoes electrochemical reduction to either ethylene glycol or methanol. The localized concentration of free formaldehyde was calculated using a reaction kinetics model that encompasses the reversible hydration reaction, electrocatalytic reaction within the carbon catalyst layer, and mass transfer of the formaldehyde from the bulk electrolyte solution as shown in Figure 2b. Since the formation of free formaldehyde is an endothermic process, higher temperatures facilitate this reaction, thus promoting the electroreduction of formaldehyde to ethylene glycol. The Faradaic efficiency and current density versus applied potential at 30°C and 40°C are provided in Supplementary Figure 9.

Figure S 9| Performance of formaldehyde electroreduction on carbon under various temperatures. (a) Formaldehyde electroreduction at 30°C. (b) Formaldehyde electroreduction at 40°C.

The following paragraph is emphasized to further explain how temperature impacts formaldehyde electroreduction.

“The presence of formaldehyde in aqueous solutions is a mixture of its hydrated form, methanediol ($\text{CH}_2(\text{OH})_2$) and dissolved free formaldehyde: $\text{CH}_2\text{O}(\text{g}) + \text{H}_2\text{O}(\text{l}) \rightleftharpoons \text{CH}_2(\text{OH})_2(\text{l})$. To proceed with electroreduction of formaldehyde, methanediol needs to go through the reverse reaction to form free formaldehyde, and then the free formaldehyde is electrochemically reduced to ethylene glycol or methanol. We hypothesize that only free formaldehyde in the carbon catalyst layer can be electrocatalytically reduced. Thus, the limiting current density is determined by the concentration of free formaldehyde in the catalyst layer. To estimate the catalyst layer free formaldehyde concentration, we constructed a reaction kinetics model considering the reversible hydration reaction, electrocatalytic reaction in the carbon catalyst layer and mass transfer of the formaldehyde from the bulk electrolyte solution. (See detailed discussion in the Supplementary Information). The modeling results (Figure 3b) show that at 200 mA cm⁻² the concentrations of free formaldehyde at 50°C is 5 times higher than that at 20°C. The majority of formaldehyde is presented as hydration form of methanediol, which is in line with the literature.¹ The concentration

of free formaldehyde increases as the temperature rises because the dehydration of methanediol is an endothermic reaction. Based on the model, we conducted the formaldehyde electroreduction at elevated temperatures and found that the partial current density of ethylene glycol increased from 15.0 mA cm⁻² to 145.56 mA cm⁻² at -1.13 V vs. RHE when the temperature increased from 30°C to 50°C (Figure 3c-d, Figure S9), consistent with the modeled local free formaldehyde concentration.”

Reference:

1. Winkelman J, Voorwinde O, Ottens M, Beenackers A, Janssen L. Kinetics and chemical equilibrium of the hydration of formaldehyde. *Chemical Engineering Science* 2002, 57(19): 4067-4076.

4、 Can similar effects be achieved if platinum plates are used to replace catalysts in the experiment of methanol preparation of formaldehyde? It is suggested that the performance of platinum sheet should be supplemented. As for Figure 4c, please give the marking and local magnification of pt, otherwise it cannot be convinced that pt is in metallic state.

Reply: The porous Pt mesh was utilized as a reference to examine the performance of bulk Pt in methanol partial oxidation. As a porous substrate, the platinum mesh maintains ionic conductivity between the anode and the membrane in a membrane electrode assembly configuration. As shown in Figure S14, Pt mesh also exhibits activity in methanol partial oxidation to formaldehyde with up to 33% FE towards formaldehyde production. However, it requires high potential (>3 V vs. RHE) to achieve a current density of 5 mA cm⁻².

The subsequent text has been incorporated into the manuscript to elucidate the catalytic activity of the platinum mesh as a reference.

“The platinum mesh also shows activity in the partial oxidation of methanol to formaldehyde, with a Faradaic efficiency of up to 33% for formaldehyde production. However, a high potential exceeding 3 V versus RHE is required to attain a current density of 5 mA cm⁻².”

Figure S 1|Performance of methanol partial oxidation to formaldehyde on Pt mesh.

The local magnification of XANES spectra for Pt catalysts under methanol oxidation reaction condition is given in the updated Figure 4c and is compared with Pt(0) and Pt(II) standard in Figure S16. The results suggest that Pt catalyst mainly maintains its metallic state under the reaction condition. Notably, an increase in the intensity of the white-line in the Pt L3-edge spectra is observed with rising current density, suggesting that platinum exhibits a more oxidized characteristic under higher potentials.¹

Figure 4c| In-situ X-ray absorption near edge structure (XANES) spectra of Pt microparticles under the methanol electrochemical partial oxidation reaction condition in anhydrous methanol containing 1 M sodium acetate.

Figure S16| The X-ray absorption near edge structure (XANES) spectra of Pt microparticles under the methanol electrochemical partial oxidation reaction condition compared with Pt(0) and Pt(II) standard.

The following context has been added into the manuscript to discuss the oxidation state of Pt catalysts under the methanol oxidation reaction conditions.

“In-situ X-ray absorption spectroscopy is applied to investigate the oxidation state of platinum in real-time during the methanol oxidation reaction conducted in anhydrous methanol containing 1 M sodium acetate as supporting electrolyte. A constant current density of 10 mA cm⁻² and 20 mA cm⁻² was applied, respectively. The in-situ X-ray absorption near edge structure (XANES) spectra of Pt catalysts was compared with Pt(0) and Pt(II) standard (Figure S16), suggesting that Pt mainly remains the metallic state under the methanol partial oxidation reaction. Notably, an increase in the intensity of the white-line in the Pt L3-edge spectra is observed with rising current density (Figure 4c), suggesting that platinum exhibits a more oxidized characteristic under higher potentials.”¹

Reference:

1. Morris DJ, Zhang P. In situ X-ray Absorption Spectroscopy of Platinum Electrocatalysts. *Chemistry–Methods* 2021, **1**(3): 162-172.

5、 The effect of the membrane electrode experiment is not well. Can we change to series system or use H-cell for two reactions at the same time?

Reply: It is feasible to change to a series system or conduct the experiment using an H cell. In this study, the reduction reaction of formaldehyde to ethylene glycol is coupled with methanol oxidation to formaldehyde in a single electrolyzer. This arrangement could eliminate the need for paired hydrogen and oxygen evolution reactions, simplifying the design and reducing the capital costs, as only one electrolyzer is required instead of two.

The reason we chose the membrane electrode assembly-based electrolyzer is to lower the internal resistance and improve the energy efficiency. In this configuration, the membrane serves as a solid electrolyte, eliminating the need for supporting electrolyte dissolved in organic solvent, which typically exhibits poor conductivity and high internal resistance. In a two-compartment flow electrolyzer configuration, even a thin layer of organic electrolyte (~2 mm) between the catalyst and membrane contributes significantly to ohmic loss, resulting in a full cell voltage of approximately 7.2 V at 100 mA cm⁻². By contrast, the MEA-based electrolyzer design, as depicted in Figure 5a, involves hot-pressing the catalyst layer onto the ionically conductive membrane. This method significantly lowers the internal resistance to 0.694 Ω compared with 5.392 Ω in the two-compartment flow electrolyzer. Consequently, the full cell voltage in the MEA-based electrolyzer is reduced to 3.2 V at 100 mA cm⁻² (Figure 5b). In the case of a traditional H-cell, the cell voltage exceeds 10 V at a low current density (less than 50 mA cm⁻²). Thus, while theoretically possible, using an H-cell may not be beneficial in terms of efficiency and cost-effectiveness.

The following paragraph is emphasized to further explain the role of membrane-electrode-assembly electrolyzer.

“The feasibility of ethylene glycol electrosynthesis directly from methanol is investigated by coupling anodic methanol partial oxidation with formaldehyde electroreduction in a single membrane electrode assembly electrolyzer. Although both methanol partial oxidation and formaldehyde electroreduction show a reasonable half-cell potential in the three-electrode configuration after iR compensation, the full cell voltage is extremely high due to the high internal resistance caused by poor solubility and conductivity of supporting electrolyte in organic phase. The conductivity of 1 molar sodium acetate in 37 wt% formaldehyde solution is 14.27 μS cm⁻¹, approximately 5 times lower than that in water. In two-compartment flow electrolyzer configuration, even a thin layer of organic electrolyte of ~ 2 mm between catalyst and membrane accounts for significant ohmic loss, resulting in a full cell voltage of ~7.2 V at 100 mA cm⁻². By using a membrane electrode assembly-based electrolyzer as shown in Figure 5a, catalyst layer is hot-pressed onto the ionic conductive membrane and the internal resistance is lowered to 0.694 Ω, compared with 5.392 Ω in two-compartment flow electrolyzer. As a result, the full cell voltage in MEA-based electrolyzer is reduced to 3.2 V at 100 mA cm⁻² (Figure 5b). The techno-economic assessment suggests that operating cell voltage plays a significant role in total cost of ethylene glycol production. Reducing the cell voltage to 3.2 V enables ethylene glycol electrosynthesis from formaldehyde has a competitive price compared with conventional manufacturing route (Figure 5c).”

Reviewer #2 (Remarks to the Author):

In the manuscript entitled: " Electrosynthesis of Ethylene Glycol from C1 Feedstocks in a Flow Electrolyzer" the authors has reported an electrochemical approach for the selective synthesis of ethylene glycol from methanol. The topic has been conveniently introduced, and it is very relevant not only academically, but also industrially as well as timely. The obtained conclusions have been adequately supported by experimental and theoretical results. Advanced characterization techniques, including in situ and in operando spectroscopic techniques, have been used to disclose the reaction mechanism, which results have been conveniently supported by theoretical calculations. Overall, those are very interesting and notable results, the research is properly driven and the manuscript is well written and easy to follow. However, there is a main issue related with the nature of the actual active sites in both electrodes that, in my opinion, remains unclear.

In one hand, the authors have demonstrated that after HCL and thermal treatment the carbon black catalyst still has residual O functional groups. Which is the actual O content? have the authors quantify? Moreover, the author has prepared two different catalysts with a higher O content, prepared by two different routes, and exhibiting different oxygenated functional groups. These catalysts have resulted in lower FE or performance than the original one, ruling out the O functional groups as active sites. However, it is still unclear which are active sites. From theoretical calculations, the graphenic basal planes seems to be more active than edges. Then, can the authors further demonstrate this experimentally using pure, ideal graphene or, alternatively, with a further reduced carbon black?? So, instead of oxidizing, as they have already done, trying to remove as much oxygenated moieties as possible. On the other hand, it seems clear that Pt planes are more active for MeOH oxidation than the edges. But, can the authors comment on the activity of the different facets?? is there any preferential orientation in the Pt microparticles?? it can be elucidated which could be the most active facet experimentally using single crystals or theoretically?

In conclusion, I recommend the publication of the report in this journal after careful revision of these issues.

Reply: We thank the reviewer for the supporting this work and providing constructive suggestions. Below is our response to the comments.

1. In one hand, the authors have demonstrated that after HCL and thermal treatment the carbon black catalyst still has residual O functional groups. Which is the actual O content? have the authors quantify? Moreover, the author has prepared two different catalysts with a higher O content, prepared by two different routes, and exhibiting different oxygenated functional groups. These catalysts have resulted in lower FE or performance than the original one, ruling out the O functional groups as active sites. However, it is still unclear which are active sites. From theoretical calculations, the graphenic basal planes seem to be more active than edges. Then, can the authors further demonstrate this experimentally using pure, ideal graphene or, alternatively, with a further reduced carbon black?? So, instead of oxidizing, as they have already done, trying to remove as much oxygenated moieties as possible.

Reply: We thank the reviewer for the insightful suggestion. The oxygen-containing functional group in carbon black catalysts can be further reduced by thermal treatment at elevated temperatures. The relationship between temperature and oxygen content is depicted in Figure S5. When subjected to a calcination process at 500°C under a 5% H₂/Ar atmosphere, the oxygen content in carbon black falls from 7.3% to 3.6%. As the thermal treatment temperature rises from 500°C to 800°C, the oxygen content further

reduces to 2.9%. We tested carbon black catalysts, with varying oxygen contents, for formaldehyde electroreduction under identical conditions. Our results indicate that by eliminating oxygen functional groups via thermal treatment, the Faradaic efficiency (FE) for ethylene glycol improved, and side reactions were minimized. However, extended annealing at higher temperatures was unable to remove additional oxygen functional groups and led to a degradation in the pore properties and surface area of the carbon black, as documented in earlier research.¹ A similar phenomenon was also noticed when using commercial graphene. Thermal treatment was unable to fully eliminate the oxygen-containing functional groups on the graphene, resulting in a residual oxygen content of 2.3% post-heat treatment. This can be attributed to the fact that the edge surface of graphene tends to remain completely oxygenated, as stated in previous literature.²

Figure S5| Thermal treatment effects on the removal of oxygen-containing functional groups on carbon black catalysts and the subsequent formaldehyde electroreduction performance. (a) The oxygen content of carbon black catalysts after undergoing thermal treatment at 500°C and 800°C, respectively. Comparative performance of formaldehyde electroreduction using carbon black catalysts without thermal treatment (b), and those treated at 500°C (c) and 800°C (d).

The following paragraph has been included to investigate the carbon black with further reduced oxygen-containing functional groups.

“The oxygen-containing functional group in carbon black catalysts can be further reduced by thermal treatment at elevated temperatures. The relationship between temperature and oxygen content is depicted in Figure S5. When subjected to a calcination process at 500°C under a 5% H₂/Ar atmosphere, the oxygen content in carbon black falls from 7.3% to 3.6%. As the thermal treatment temperature rises from 500°C to 800°C, the oxygen content further reduces to 2.9%. We tested carbon black catalysts, with varying oxygen contents, for formaldehyde electroreduction

under identical conditions. Our results indicate that by eliminating oxygen functional groups via thermal treatment, the Faradaic efficiency (FE) for ethylene glycol improved, and side reactions were minimized. However, extended annealing at higher temperatures was unable to remove additional oxygen functional groups and led to a degradation in the pore properties and surface area of the carbon black, as documented in earlier research.¹ A similar phenomenon was also noticed when using commercial graphene. Thermal treatment was unable to fully eliminate the oxygen-containing functional groups on the graphene, resulting in a residual oxygen content of 2.3% post-heat treatment. This can be attributed to the fact that the edge surface of graphene tends to remain completely oxygenated, as stated in previous literature.²”

Reference:

1. Kim J-H, Kim S-H, Kim B-J, Lee H-M. Effects of Oxygen-Containing Functional Groups on the Electrochemical Performance of Activated Carbon for EDLCs. *Nanomaterials* 2023, 13(2): 262.
2. Intan NN, Pfaendtner J. Composition of Oxygen Functional Groups on Graphite Surfaces. *The Journal of Physical Chemistry C* 2022, 126(26): 10653-10667.

2. On the other hand, it seems clear that Pt planes are more active for MeOH oxidation than the edges. But, can the authors comment on the activity of the different facets?? is there any preferential orientation in the Pt microparticles?? it can be elucidated which could be the most active facet experimentally using single crystals or theoretically?

Reply: We thank the reviewer for the insightful suggestion. We acknowledge that the study of the most active facet, either experimentally using single crystals or theoretically, could provide valuable insights into the catalytic methanol oxidation process. However, an in-depth examination of these aspects is beyond the scope of our current study. The primary objective of this paper is to provide a proof-of-concept for the electrosynthesis of ethylene glycol from methanol. We anticipate that further work will be conducted in the future to identify the preferential facet of Pt for catalyzing the oxidation of methanol to formaldehyde.

Reviewer #3 (Remarks to the Author):

The manuscript describes a very interesting alternative for the synthesis of ethylene glycol, a very important chemical in the chemical industry and responsible for the significant production of CO₂ emissions. The authors suggest synthesizing ethylene glycol from methanol via formaldehyde. This process will happen in two steps: methanol incomplete oxidation to formaldehyde on a Pt-based electrode and formaldehyde reduction to ethylene glycol on a carbon-based catalyst. In addition to the novelty and scientific relevance, the manuscript is extremely complete. The authors investigate the reaction mechanism by using labelled compounds, DFT and operando analytics in addition to the optimized reaction conditions (pH, temperature and concentration). My main concern with the manuscript, being a communication, is the amount of information in the main text. For example, the investigation of the active site of carbon-based material can be shortened in the main manuscript and moved to the SI. Moreover, I would suggest presenting the results for the Reaction mechanism of C-C coupling together with the Formaldehyde electroreduction to ethylene glycol in a summarized manner. Strategies like this will improve the readability and understanding of the paper.

In addition, I would like the authors to address the following general points.

Reply: We appreciate the reviewer's endorsement of our work. In response to the comments, we have relocated some of the investigation regarding the active site on carbon to the supporting information. Please find our detailed responses to the comments below.

1- Can the authors comment on the impact of the origin of methanol on the overall sustainability of the process?

Reply: We appreciate the reviewer's comment on the impact of methanol origin on the sustainability of our ethylene glycol electrosynthesis process. Although methanol is predominantly produced from syngas derived from fossil fuels, there has been growing interest and progress in developing more sustainable methods, such as CO₂ hydrogenation. The world's first commercial-scale CO₂-to-methanol plant in China exemplifies this approach, capturing 160,000 tons of CO₂ from air and producing 110,000 tons of methanol annually. By utilizing CO₂ as a carbon source, this method contributes to greenhouse gas emissions reduction and provides a sustainable source of methanol. Consequently, we propose coupling our ethylene glycol electrosynthesis process with methanol production from CO₂ hydrogenation to enhance the overall sustainability of the process. This integration establishes a closed-loop system, wherein ethylene glycol is produced from CO₂, promoting renewable carbon sources, and improving overall sustainability.

The following context has been added to the updated manuscript highlighting the influence of methanol's source on the overall sustainability of the process.

“This process can be integrated with the emerging CO₂ hydrogenation-based methanol production to improve the overall sustainability, showcasing a sustainable route from CO₂ to ethylene glycol.”

2- Can the authors comment on the long-term stability of the electrolyzer? I that formaldehyde transport through the existent comments membranes and the use of organic solvents will not be optimal for the stability of the electrolysis.

Reply: In our study, we conducted a 10-hour stability test, during which we did not observe any degradation of the Nafion membrane caused by formaldehyde or methanol. We acknowledge that membrane stability is a crucial factor in the long-term performance of the membrane electrode assembly (MEA) configuration.

As such, we plan to focus more on membrane stability in future research, exploring the performance and durability of the membrane under various conditions within the MEA configuration.

3- The authors should provide further information on the external reference electrode used and how they overcome the issues of the organic solvents.

Reply: Ag/AgCl (Pine research) was used as the external reference electrode and was calibrated against a brand-new Ag/AgCl electrode every time before and after the experiment using a voltage meter. No obvious change in the potential was observed during short-term use for formaldehyde electroreduction.

The following statement has been incorporated into the manuscript in response to the reviewer's comment.

“An Ag/AgCl (Pine Research) external reference electrode was employed and calibrated against a new Ag/AgCl electrode before and after each experiment using a voltage meter. No significant potential changes were observed during short-term use for formaldehyde electroreduction.”

4- Can the authors comment on the use of Pt-based electrodes, when all the scientific community tries to avoid the use of PGM?

Reply: We acknowledge the need to minimize PGM catalyst usage, and potential solutions include using PGM-free catalysts or introducing a second metal to lower the loading of Pt. In direct methanol oxidation fuel cell studies, research has explored non-precious metal alternatives, such as Ni-based catalysts.^{1,2} Furthermore, studies have investigated introducing secondary metals like Sn to reduce Pt loading.³ Our work presents a proof of concept for methanol partial oxidation to formaldehyde using Pt catalysts and coupling it with formaldehyde electroreduction to ethylene glycol. Future work will focus on exploring PGM-free catalysts for methanol partial oxidation to selectively produce formaldehyde.

The subsequent statement has been added to the manuscript, emphasizing the necessity for Platinum Group Metal (PGM)-free catalysts.

“Our proof-of-concept study employed Pt catalysts for methanol partial oxidation to generate formaldehyde. Future research will focus on exploring the use of Platinum Group Metal (PGM)-free catalysts or incorporating secondary metals, with the objective of reducing Pt usage.”

Reference:

1. Abdelkareem MA, Sayed ET, Mohamed HO, Obaid M, Rezk H, Chae K-J. Nonprecious anodic catalysts for low-molecular-hydrocarbon fuel cells: Theoretical consideration and current progress. *Progress in Energy and Combustion Science* 2020, 77: 100805.
2. Durai L, Gopalakrishnan A, Badhulika S. Highly Stable NiCoZn Ternary Mixed-Metal-Oxide Nanorods as a Low-Cost, Non-Noble Electrocatalyst for Methanol Electro-Oxidation in Alkaline Medium. *Energy & Fuels* 2021, 35(15): 12507-12515.
3. Hasa B, Martino E, Tsatsos S, Vakros J, Kyriakou G, Katsaounis A. Non-precious Sn as alternative substitute metal in graphene-based catalysts for methanol electrooxidation. *Journal of Applied Electrochemistry* 2022, 52(3): 509-520.

REVIEWERS' COMMENTS

Reviewer #1 (Remarks to the Author):

I have carefully examined the response from the authors. They have well addressed my concerns and thus this work can be published.

Reviewer #2 (Remarks to the Author):

In my opinion, the authors have successfully answered all issues. The overall quality of the manuscript have been improved through implementation of referees's suggestions and clarifications. For that reasons, I recommend for publication in the present form.